# Less Severe Polymicrobial Sepsis in Conditional *mgmt*-Deleted Mice Using LysM-Cre System, Impacts of DNA Methylation and MGMT Inhibitor in Sepsis

**DOI:** 10.3390/ijms241210175

**Published:** 2023-06-15

**Authors:** Kritsanawan Sae-khow, Pornpimol Phuengmaung, Jiraphorn Issara-Amphorn, Jiradej Makjaroen, Peerapat Visitchanakun, Atsadang Boonmee, Salisa Benjaskulluecha, Tanapat Palaga, Asada Leelahavanichkul

**Affiliations:** 1Medical Microbiology, Interdisciplinary and International Program, Graduate School, Chulalongkorn University, Bangkok 10330, Thailand; kritsanawan_29@hotmail.com; 2Department of Microbiology, Faculty of Medicine, Chulalongkorn University, Bangkok 10330, Thailand; pphuengmaung@gmail.com (P.P.); peerapat.visitchanakun@gmail.com (P.V.); 3Center of Excellence in Translational Research in Inflammation and Immunology (CETRII), Faculty of Medicine, Chulalongkorn University, Bangkok 10330, Thailand; jiraphorn298@gmail.com; 4Center of Excellence in Systems Biology, Research Affairs, Faculty of Medicine, Chulalongkorn University, Bangkok 10330, Thailand; jiradejmak@gmail.com; 5Department of Microbiology, Faculty of Science, Chulalongkorn University, Bangkok 10330, Thailand; atsadang88@gmail.com (A.B.); salisafaii@gmail.com (S.B.); tanapat.p@chula.ac.th (T.P.); 6Division of Nephrology, Department of Medicine, Faculty of Medicine, Chulalongkorn University, Bangkok 10330, Thailand

**Keywords:** sepsis, lipopolysaccharide, macrophages, epigenetics, *mgmt*

## Abstract

The O6-methylguanine-DNA methyltransferase (MGMT) is a DNA suicide repair enzyme that might be important during sepsis but has never been explored. Then, the proteomic analysis of lipopolysaccharide (LPS)-stimulated wild-type (WT) macrophages increased proteasome proteins and reduced oxidative phosphorylation proteins compared with control, possibly related to cell injury. With LPS stimulation, *mgmt* null (*mgmt*^flox/flox^; LysM-Cre^cre/-^) macrophages demonstrated less profound inflammation; supernatant cytokines (TNF-α, IL-6, and IL-10) and pro-inflammatory genes (*iNOS* and *IL-1β*), with higher DNA break (phosphohistone H2AX) and cell-free DNA, but not malondialdehyde (the oxidative stress), compared with the littermate control (*mgmt*^flox/flox^; LysM-Cre^-/-^). In parallel, *mgmt* null mice (MGMT loss only in the myeloid cells) demonstrated less severe sepsis in the cecal ligation and puncture (CLP) model (with antibiotics), as indicated by survival and other parameters compared with sepsis in the littermate control. The *mgmt* null protective effect was lost in CLP mice without antibiotics, highlighting the importance of microbial control during sepsis immune modulation. However, an MGMT inhibitor in CLP with antibiotics in WT mice attenuated serum cytokines but not mortality, requiring further studies. In conclusion, an absence of *mgmt* in macrophages resulted in less severe CLP sepsis, implying a possible influence of guanine DNA methylation and repair in macrophages during sepsis.

## 1. Introduction

The imbalanced immune responses in patients with sepsis, a potentially life-threatening condition from severe infection regardless of the organisms, result in severe hyperinflammation, despite effective microbial control by antibiotics [1,2,3]. Although there is improved supportive care in sepsis [4], the inflammatory blockage during sepsis-hyperinflammation [5,6,7,8,9,10,11,12] may still help reduce sepsis mortality. Among several sepsis pro-inflammatory factors, the presence of LPS (a major cell wall component of Gram-negative bacteria) in the blood, referred to as endotoxemia [13,14,15,16], possibly due to Gram-negative bacteremia or translocation of LPS from the gut into the blood circulation (leaky gut) [17,18,19] is a well-known cause. Interestingly, monocytes (or macrophages) are important cells controlling microbial molecules (LPS and other substances) [20,21], and the responses to LPS lead to several cell activities, including epigenetic modifications, chromatin remodeling, and interferences on cell energy status [22,23,24]. Epigenetics, the phenotypic alterations without the changes in the DNA sequence [25,26] used for the switch-on and -off DNA transcription through the modifications on DNA and histone (methylation and acetylation) or with the noncoding RNA (microRNA) [27], are one of the interesting macrophage responses after LPS activation. Indeed, the modifications of DNA and histone rely on several enzymes to modify the physiological outcomes [28]. DNA methylation, among all modifications, for regulating DNA repair is a fascinating mechanism in LPS responses. Because any forms of oxidative stress from both regular procedures or pathogenic processes (such as LPS stimulation and exposure to alkylating agents) in the cells can induce DNA modifications, DNA methylation is a common process regularly found in every body cell [29]. Indeed, DNA methylation, especially the N-methylated purines and O6-methylguanine (O6MeG), are common DNA damage that can trigger point mutation with high mutagenicity and carcinogenicity [30]. The O6MeG, which the alkylating agent usually activates, is mutagenic because polymerase enzymes typically mismatch insert thymine instead of cytosine (O6MeG:T) due to the similar strength of the hydrogen bonds to cytosine and thymine [31]. Not only alkylating agents and environmental compounds but several endogenous factors, particularly oxidative stress, are also responsible for producing O6MeG during routine cell activities [32]. Base damage, single-strand breaks (SSB), double-strand breaks (DSB), and inter-strand cross-links are all types of DNA damage caused by normal metabolic processes (hydrolysis, deamination, alkylation, and oxidation) which occur about 50,000 lesions per cell per day (or about 30,000 nucleoside sites in DNA per cell) [33]. Although the number of modifications at the O6 position of guanine (O6MeG) may not be significantly high compared to the methylation in the total lesions on DNA [34], the amount of O6MeG may be increased in sepsis or LPS hyper-responsiveness due to high levels of oxidative stress as indirectly indicated by inflammation-induced cancer in some situations [35]. Interestingly, O6MeG may result in more severe DNA damage and cell death than methylation at the other DNA locations due to the easier point mutation and more potent DNA damage [36], which might also profoundly affect macrophages than other mechanisms of epigenetic changes. 

To maintain genome stability, DNA repair is necessary, partly through the removal of methyl groups, on the DNA by base excision repair initiated by the alkyladenine-DNA glycosylase, the family of alkylation B (AlkB) homologs proteins, and the suicidal enzyme O6-methylguanine-DNA methyltransferase (MGMT) [30,37]. Following DNA repair, these enzymes’ methylated and alkylated forms are rapidly broken down. Intact MGMT prevents the malignant transformation of various tissues. MGMT blocking is employed for adjuvant chemotherapy [30] in malignancies that depend on the rate of MGMT re-synthesis in the cancer cells [37]. The DNA methylation and damage in the activated macrophages are conceivable because of several activator signals and immune activation-induced oxidative stress [38], particularly after induction with LPS, a microbial molecule with an inflammatory-inducing solid property. Indeed, DNA methylation in macrophages is directly induced by LPS [39] and indirectly activated by reactive oxygen species (ROS) generated by LPS responses [40]. In patients with sepsis, higher levels of genomic DNA hypermethylation patterns are linked to pro-inflammatory pathways [41]. The effects of this methylation and MGMT on macrophages, the cells with a potentially high stress-induced DNA methylation [42], are still unknown, despite a strong conclusion on the involvement of O6MeG in malignant cells. Accordingly, sepsis and LPS cause DNA methylation in several sites, perhaps including Q6MeG [41], and the MGMT enzyme reduces O6MeG during the DNA repair that revitalizes the cell functions. Then, the MGMT inhibitors (such as Lomeguatrib) enhance the death of cancer cells through interference with DNA repair [43], and this interference on macrophages during sepsis may reduce macrophage hyper-inflammation [44]. Additionally, the screening of epigenetic inhibitors reveals that MGMT inhibitors change the expression of inflammatory cytokines in LPS-activated macrophages [45], and the MGMT inhibitors not only neutralize O6MeG in DNA but also link to the repair of other pathways [46]. Despite several active research areas, controlling macrophage responses through epigenetic manipulations is an exciting method for managing immune responses during sepsis [10,47].

Then, we hypothesized that failure of DNA repair causes a reduction in macrophage functions, especially cytokine production, that might attenuate sepsis-induced hyper-inflammatory responses. Here, we explored the impact of *mgmt* on macrophage responses to LPS and cecal ligation and puncture (CLP) sepsis model using the conditional *mgmt* deletion mice with LysM-Cre system that selectively affected *mgmt* only in myeloid cells. 

## 2. Results

### 2.1. Proteomic Analysis of Lipopolysaccharide (LPS)-Activated Wild-Type Macrophages and Impacts of mgmt Null and MGMT Inhibitor on LPS Stimulation

The alteration of wild-type (WT) macrophages after LPS stimulation was explored by proteomic analysis through the list of the genes that generated proteins, enrichment pathway, and KEGG analysis (Figure 1A–C and Figure 2A–C). There were proteins from 119 and 206 up-and down-regulated genes in LPS-activated macrophages compared with the control (Figure 1A and Figure 2A). Functions of the elevated proteins in LPS-stimulated cells compared with the control, as indicated by the enrichment analysis, involved in the proteasome, carbohydrate metabolism, antigen presentation, and several infections (Figure 1B). The KEGG analysis of these LPS-enhanced proteins in proteasome was demonstrated in Figure 1C, and the genes that appeared in the analysis were Rpn3, Rpn6, Rpn7, Rpn8, Rpn9, PA28α, and PA28β (Figure 1C; red color boxes). Notably, the RPN is a subunit of the 19S regulatory particle (RP) at the lid portion of the 26S proteasome [48], while PA28α and PA28β are proteasome activators forming a heteropolymer that binds to both ends of the 20S proteasome, referred to as “immunoproteasome”, using for the processing of certain antigens [49]. In parallel, functions of the LPS-decreased proteins in macrophages by the enrichment analysis involved in cell energy (citrate cycle and oxidative phosphorylation), synthesis of protein and lipid, and several diseases (Figure 2B). Additionally, the KEGG analysis of these LPS-suppressed proteins in the oxidative phosphorylation, a significant mechanism for providing cell energy, was demonstrated in Figure 2C, including several Ndufs (NADH: Ubiquinone Oxidoreductase Core Subunits) of NADH (nicotinamide adenine dinucleotide + hydrogen) dehydrogenases [50], a component of cytochrome C reductase and oxidase, consisting of a few QCRs (quinol-cytochrome c oxidoreductases) and COX5B (cytochrome c oxidase subunit 5B) [51,52], with several parts of F-type ATPase, such as OSCP (oligomycin-sensitivity-conferring protein) [53]. The list and details of the genes generating the up and down- proteins from LPS-activated macrophages using the GO enrichment analysis were demonstrated in Appendix A.

Because the reduced cell energy status (mitochondrial oxidative phosphorylation) possibly leads to abnormal molecules and proteins degraded by the proteasome [54,55], the decreased proteins in oxidative phosphorylation and increased proteins of proteasome might be due to LPS-induced cell injury [56,57,58], partly including the DNA methylation. Then, the impact of LPS activation was further explored in macrophages from mgmt null and littermate control (mgmt control) mice. Indeed, mgmt null macrophages demonstrated lower inflammatory responses as indicated by supernatant cytokines (TNF-α, IL-6, and IL-10) and expression of pro-inflammatory M1 macrophage polarization genes, including nitric oxide synthase (iNOS) and interleukin-1β (IL-1β) without an impact on M2 polarization genes; arginase-1 (Arg-1), transforming growth factor-β (TGF-β), and resistin-like molecule-1 (Fizz-1) (Figure 3A–H). Likewise, administration of the MGMT inhibitor in LPS-activated wild-type (WT) macrophages also demonstrated anti-inflammatory effects through these parameters (except for supernatant IL-6) (Figure 3I–P), suggesting a possible use of MGMT inhibitor to attenuate sepsis-induced hyperinflammation. These data also imply the reduced macrophage cytokine production, perhaps due to methylation of the DNA that is responsible for the production of these cytokines, due to the lack of MGMT to repair the DNA [42]. Because mgmt null macrophages might demonstrate higher cell injury than the control cells due to persistent DNA methylation from the loss of MGMT enzymes, several injury parameters were evaluated. As such, mgmt null macrophages demonstrated higher supernatant cell-free DNA and the DNA break (phosphohistone H2A.X) but not cell reactive oxygen species (evaluated by Malondialdehyde; MDA) (Figure 4A–D). Notably, phosphorylation of the histone variant H2AX is a critical factor for DNA damage response to assembly of the DNA repair proteins at the chromatin damaged sites [59], and MDA is the final peroxidation products of polyunsaturated fatty acids activated by several inducers, including LPS [60]. Perhaps, the loss of MGMT enzyme leads to more profound DNA damage without repair, as indicated by the DNA break indicator (Figure 4C,D), which results in more potent cell injury (cell-free DNA) (Figure 4A) but does not affect lipid peroxidation injury (MDA) (Figure 4B).

### 2.2. The Mgmt Null Mice Demonstrated Less Severe Cecal Ligation and Puncture (CLP) Sepsis than the Littermate Control

Because of the impacts of mgmt-manipulated macrophages (Figure 3 and Figure 4) and the importance of antibiotics in sepsis [61], further exploration in mgmt null mice (mgmt^fl/fl^; LysM-Cre^cre/-^) and mgmt littermate control (mgmt^fl/fl^; LysM-Cre^-/-^) using CLP surgery with and without antibiotics were performed. With antibiotic use, mgmt null mice demonstrated less severe sepsis than sepsis in littermate control, as indicated by survival analysis, kidney injury (serum creatinine and renal histology score), liver damage (serum alanine transaminase and liver histological score), spleen apoptosis, cell-free DNA, endotoxemia, bacteremia, and serum cytokines (TNF-α, IL-6, and IL-10) (Figure 5A–G and Figure 6A–G). However, the protective effect against sepsis of mgmt null mice was lost in CLP without antibiotics as indicated by survival analysis, serum creatinine, alanine transaminase, and serum cytokines (TNF-α, IL-6, and IL-10) (Figure 7A–F) supporting the necessity of microbial control during immune modulation in sepsis [62]. For clinical translation purposes, Lomeguatrib (an MGMT inhibitor) was further tested in WT mice with an antibiotic-administered CLP model. Although the inhibitor could not attenuate mortality and organ injury (kidney and liver), the serum cytokines of the treated mice were lower than the control mice (Figure 7G–L). Notably, there was no kidney and liver injury in sham mice with the inhibitor, implying less toxicity to the kidney and liver of the inhibitor. 

## 3. Discussion

### 3.1. Possible LPS-Induced Macrophage Injury from the Proteomic Analysis in the Wild-Type Cells

The presence of lipopolysaccharide (LPS) in blood circulation (endotoxemia) can be found in several conditions [63,64,65], predominantly due to gut barrier damage [3,18,20,66] and Gram-negative bacteremia [2,3,67]. As such, LPS (one of the pathogen-associated molecular patterns; PAPMs) stimulates all cells in the body, including immune cells, and macrophages are the major cells responsible for recognising and controlling most foreign molecules, including LPS [68,69]. Although only the high abundance proteins are detectable by proteomic analysis, evaluating these proteins might crudely identify the direction of cell response. Here, we demonstrated a higher number of down-regulated proteins than the up-regulated groups in LPS-stimulated macrophages, as listed in Appendix A. With enrichment pathway analysis, the upregulated proteins correlated with the proteasome, the protein complexes used for degrading the damaged proteins by proteolysis with several protease enzymes [70,71]. Despite several functions of the proteasome in cell homeostasis, the 26S proteasome regulates DNA repair by degrading the proteins or acting as a molecular chaperone to promote the disassembly of the repair complex [54]. Increased proteasome in LPS-activated macrophages indirectly indicates an increased abundance of unneeded proteins after LPS stimulation supporting previous publications [72,73].

On the other hand, enrichment pathways of the down-regulated proteins were correlated with cell metabolisms and energy status, including 2-oxocarboxylic acids (association with pyruvate pathway), citrate cycle (glycolysis), and oxidative phosphorylation (mitochondrial function). Perhaps, the increased cell energy utilization during macrophage responses to LPS (cytokine production, inflammatory signaling synthesis, etc.) uses up these proteins that partly be associated with mitochondrial injury and increased reactive oxygen species [74]. Although proteins for the modulation of DNA and histone were not found in the proteome results, possibly due to too little abundance of these proteins, the proteomic analysis indirectly supported a possible cell injury, including LPS-induced DNA methylation as previously published [57].

The DNA methylation is more frequently demonstrated at the cytosine site than the guanine position, especially at the cytosine-phosphate-guanine (CpG), as more than 70–80% of CpG sites in humans are modified, despite the lower abundance of CpG in humans compared with prokaryotic cells [75,76]. Both DNA and histone alterations are critical regulators of gene expression through the chromatin structures [77], using several key enzymes to control the chromatin accessibility that is well-known in cancer [78] but has fewer data in sepsis [79]. For DNA methylation, it is the transfer of a methyl group, frequently to the C-5 position of the cytosine ring of DNA-by-DNA methyltransferase (DNMT) in any cytosines of the genome, especially at the CpG regions [80]. In previous publications, the enhanced DNMT activity increases DNA methylation and aggravates pro-inflammatory macrophages and the *dnmt1* deletion of enhanced anti-inflammation [42] and attenuates macrophage inflammatory responses [81]. Despite the common cytosine methylation at CpG sites [82], methylation of guanine at the O-6 positions (O6MeG) that are controlled by O6-methylguanine-DNA methyltransferase (MGMT) caused genotoxicity [83,84] as the insufficient MGMT worsens cell injury [85].

Interestingly, DNA methylation impairs DNA transcription and induces programmed cell death, especially apoptosis [86]. Because (i) previous studies of DNA methylation in sepsis are mentioned [41,87,88,89], (ii) the possibility that increased O6MeG (due to the loss of *mgmt* for DNA repair) might enhance cell injury [85], (iii) the availability of *mgmt* inhibitor for anti-cancer [90] that possibly be helpful for sepsis [91], and (iv) epigenetic changes and in vitro tests of *mgmt* inhibitors in LPS-activated macrophages [45,92], further tests on mice with the depletion of MGMT enzyme only in the myeloid cells are interesting. Theoretically, MGMT deficiency should interfere with the macrophage activities leading to an anti-inflammatory direction which might be beneficial for treating sepsis hyper-inflammation [93,94,95]. Then, further tests on *mgmt* null macrophages and mice were performed. 

### 3.2. Impact of mgmt and DNA Repair in Sepsis, In Vitro and In Vivo

To test the impacts of MGMT enzyme in sepsis, bone marrow-derived macrophages from *mgmt* null mice (*mgmt^fl/fl^; LysM-Cre^cre/-^*) and an MGMT inhibitor were used. As such, *mgmt* null macrophages and MGMT inhibitors induce anti-inflammatory effects as indicated by reduced supernatant cytokines and the down-regulated genes of M1 pro-inflammatory macrophages (*iNOS* and *IL-1β*) compared with the control. Due to the influence of MGMT enzyme on the DNA repairs through the removal of O6MeG (methyl group at the 6th oxygen molecule on guanine) of DNA, the deletion of MGMT in macrophages might be responsible for the enhanced cell injury as indicated here, by the higher abundance of DNA break and supernatant cell-free DNA in *mgmt* null macrophages than the control cells after LPS activation. Accordingly, phosphor-H2AX (γ-H2AX) is a marker of DNA double-strand break (DSB), a disruption of both DNA strands compromising genomic stability [96], which regularly occurs in any eukaryotic cells in the order of 10 to 50 per cell per day, depending on the cell cycles and cell types [97], due to the exogenous and endogenous inducers, including radiation, chemicals, LPS, ROS, DNA replication, and repair [98,99]. Interestingly, LPS increased ROS, and both factors (LPS and ROS) are well-known to induce DNA methylation [57,100], especially the methylation at essential sites on DNA, which is mostly mentioned as DNA break in the mutated cells [101]. In non-immune cells, several alkylating agents induce DNA methylation, especially at the guanine sites more than the cytosine positions, causing DNA mismatch binding that is toxic to the cells resulting, at least in part, in DNA break, cell mutation, or cell death [102,103]. In macrophages, LPS-induced DNA damage through LPS-mediated ROS is well-known as the damage is detectable in more than 95% of macrophages (in vitro) within only 30 min of LPS incubation, and the damage is completely protected by anti-oxidants [99]. The increased ROS and cell death in macrophages after LPS activation are also mentioned [104,105]. While DNA damage may enhance macrophage inflammation through several DNA sensors responding to the damage [106], the damage that cannot be repaired might negatively affect macrophages. Then, the interfered equilibrium between the generation of DNA break and DNA repair might be another important intervention on macrophages [40]. Here, more prominent DNA break and cell-free DNA in *mgmt* null macrophages over the control cells indirectly indicate the possible presence of DNA methylation at the guanine (O6MeG), despite the technical limitation on the direct O6MeG detection [107]. Notably, the detectable DNA damage in control untreated macrophages supports DNA injury from normal cellular processes, possibly due to stress-induced ROS, which might be more prominent in macrophages than other cell types [105,108,109]. The DNA methylation at guanine seems important in LPS-activated macrophages, perhaps due to the maintenance of cell viability and attenuation of cell injury, despite the uncommon DNA methylation at the guanine site compared with cytosine sites [80]. The extended use of an MGT inhibitor (Lomeguatrib), a chemotherapeutic agent [110], on attenuation of sepsis hyper-inflammation was also proposed following these in vitro results. 

Following the in vitro results, the cecal ligation and puncture (CLP) abdominal sepsis model was tested in the littermate control (*mgmt^fl/fl^; LysM-Cre^-/-^*) and *mgmt* null mice. As such, the CLP model is used because it is a sepsis model that more resembles human conditions than the LPS injection model, as indicated by the presence of bacteremia, cytokine levels, and other parameters [67]. Interestingly, sepsis in *mgmt* null mice was less severe than in the littermate control mice, as indicated by survival, serum cytokines, and organ injury, only with antibiotics, supporting the importance of microbial control during sepsis immunotherapy [111]. However, sepsis was not worsened in *mgmt* null mice. However, macrophages in these mice might be more susceptible to injury than the control due to an abnormality in DNA repair from the loss of the MGMT enzyme.

Interestingly, sepsis attenuation in the mice without MGMT only in myeloid cells (macrophages and neutrophils), but not other cells, implying the importance of these cells in sepsis hyper-immune responses [112] and the blockage of only the MGMT enzyme in macrophages might effectively prevent severe sepsis with fewer drug complications. However, the administration of MGMT inhibitor in CLP mice did not improve survival rate compared with the vehicle control, even with antibiotic administration, perhaps due to the drug effects on all cells in the body, not only on macrophages. Nevertheless, the MGMT inhibitor in CLP mice attenuated serum cytokines but not liver and kidney injury. Although there was no liver and kidney injury in sham mice with the MGMT inhibitor, these organs during sepsis might be more susceptible to the injury and the MGMT inhibitor effect on renal cells, and hepatocytes might be responsible for the non-improved organ injury in MGMT inhibitor-administered mice with sepsis. Perhaps, the dose adjustment, a better drug formula preparation, and/ or the selective delivery of MGMT inhibitors only on myeloid cells [113] might increase MGMT inhibitors’ effectiveness and reduce complications. Despite the necessity of more studies, we reported a proof of concept to use MGMT inhibitors, the available adjunctive anti-cancer drugs, to attenuate sepsis-induced hyper-inflammation.

### 3.3. Clinical Aspect and Future Experiments

Overall, LPS activation during sepsis induces DNA methylation in macrophages that requires some enzymes, including MGMT, to remove the methyl groups (DNA repair) to revitalize and maintain regular macrophage functions (continuous cytokine production and inflammatory response). Then, failure of DNA repair causes a reduction in macrophage functions, especially cytokine production and hyper-inflammatory responses, which possibly turns out to be a beneficial effect on sepsis. In cancer therapy, several alkylating agents destroy cancer cells through the induction of DNA methylation, and some malignant cells resist these anti-malignant drugs partly by the increased production of DNA methylation enzymes, including MGMT, to counteract the DNA damage [90]. Hence, the administration of MGMT inhibitors and some alkylating agents enhances the anti-malignant effect of some cancers, which is used as adjuvant therapy in some types of cancer [90]. In sepsis, inflammation-induced DNA methylation in macrophages seems important for the pathophysiology of sepsis-induced organ injuries (kidney, liver, and spleen). MGMT deletion only in myeloid cells (mostly macrophages and neutrophils) attenuated the injury. Indeed, infiltration of immune cells in several organs during systemic inflammation, such as sepsis and auto-immune diseases, is one of the main pathogenesis of organ injury, and the infiltration of cells with less pro-inflammatory activities by the MGMT interference here might induce less severe injury than infiltration by the very active immune cells [14,69]. Thus, the reduction of immune cell activities, including by the MGMT blockage, might be beneficial in sepsis. 

Because MGMT enzyme is an essential enzyme, not only for cancer cells but also for every cell due to the physiologic DNA methylation induced by regular cell activities, MGMT blockage might be toxic or mutagenic [110] to the cells, and the short course of treatment and/or the direct drug delivery into macrophages are interesting. Moreover, administration of MGMT blockage should be used only in a condition with a well microbial control as inflammation is necessary for organismal control, and too less inflammation might enhance secondary infection [114,115]. Therefore, although the direct detection of Q6MeG on macrophages in patients with sepsis might be the most interesting indicator for using MGMT blockage in sepsis, proper methods of O6MeG detection in patients will be needed [107,116]. For instance, high serum IL-6 and IL-1 as the hyper-inflammation markers [117,118] plus elevated Q6MeG and normal HLA-DR (a marker of sepsis immune exhaustion [114,115]) with negative blood culture (a biomarker of good control of the source of infection) might be the indicators for properly using of MGMT inhibitors in sepsis. Therefore, further experiments on MGMT inhibitors in sepsis are interesting. 

## 4. Materials and Methods

### 4.1. The Proteomic Analysis

Bone marrow-derived macrophages (BMDM) were prepared from the femurs of wild-type (WT) mice using supplemented Dulbecco’s Modified Eagle’s Medium (DMEM) with a 20% conditioned medium of the L929 cells (ATCC CCL-1) as previously described [68,119,120,121]. Macrophages at 5 × 10^4^ cells/well in supplemented DMEM (Thermo Fisher Scientific) were incubated in 5% carbon dioxide (CO_2_) at 37 °C for 24 h before being treated by lipopolysaccharide (LPS; *Escherichia coli* 026:B6) (Sigma-Aldrich, St. Louis, MO, USA) at 100 ng/mL or DMEM (control) for 24 h before in-solution digestion and peptides labeling using the light reagents (CH_2_O and NaBH_3_CN) and heavy reagents (13CD2O and NaBD3CN), respectively. The pooled peptides from macrophages in the wells were fractionated using a high pH reversed-phase peptide fractionation kit (Thermo Fisher Scientific, San Jose, CA, USA) and Liquid chromatography–tandem mass spectrometry (LC-MS/MS) was performed on an EASY-nLC1000 system coupled to a Q-Exactive Orbitrap Plus mass spectrometer equipped with a nanoelectrospray ion source (Thermo Fisher Scientific, San Jose, CA, USA). The mass spectrometry (MS) raw files were searched against the Mouse Swiss-Prot Database (17,138 proteins). In parallel, the search parameters were set for fixed modifications; carbamidomethylation of cysteine (+57.02146 Da), light and heavy dimethylation of N termini and lysine (+28.031300 and +36.075670 Da), and variable modification: oxidation of methionine (15.99491 Da). The false positive discovery rate of the identified peptides based on Q-values using The Proteome Discoverer decoy database together with the Percolator algorithm was set to 1%, and the relative MS signal intensities of dimethyl labeled peptides were quantified and presented as ratios of LPS/ control. Log 2 of the ratios in triplicate was used to calculate the *p*-values using Student’s *t*-test with a *p*-value < 0.05 as a significant difference. Then, these proteins were applied to the online DAVID Bioinformatics Resources 6.8 to investigate the enriched biological processes. The mass spectrometry proteomics data have been deposited to the ProteomeXchange Consortium via the PRIDE partner repository with the dataset identifier PXD041265. Then, the data visualization was performed using Excel and R packages. Meanwhile, KEGG pathway and Go enrichment analyses were generated by PathfindR and Shiny 0.77 (http://bioinformatics.sdstate.edu/go/, accessed on 28 May 2023), respectively.

### 4.2. The In Vitro Experiments

The BMDM from *mgmt* control (*mgmt*^fl/fl^; LysM-Cre^-/-^) or *mgmt* null (*mgmt*^fl/fl^; LysM-Cre^cre/-^) mice were extracted from mouse femurs and macrophages at 5 × 10^4^ cells/well were activated by LPS (*Escherichia coli* 026:B6) (Sigma-Aldrich) at 100 ng/mL or DMEM for 24 h. Then, supernatant cytokines (TNF-α, IL-6, and IL-10) and gene expression were measured by ELISA (Invitrogen, Carlsbad, CA, USA) and quantitative real-time polymerase chain reaction (PCR), respectively, as previously described [87]. In brief, TRIzol Reagent (Invitrogen, Carlsbad, CA, USA) together with RNeasy Mini Kit (Qiagen, Hilden, Germany) was used to extract RNA from the samples, and 1 mg of total RNA was used for cDNA synthesis with iScriptreverse transcription supermix (Bio-Rad, Hercules, CA, USA) on a QuantStudio 5 real-time PCR system (Thermo Fisher Scientific) using SsoAdvanced Universal SYBR Green Supermix (Bio-Rad, Hercules, CA, USA). The gene expressed values normalized by Beta-actin (β-actin; an endogenous housekeeping gene) with the calculated fold change by the ∆∆Ct method was conducted with primers listed in Table 1. In parallel, BMDM of WT mice were used. As such, WT BMDM at 5 × 10^4^ cells/well were activated with LPS (100 ng/mL) or DMEM with or without MGMT inhibitor (Lomeguatrib) (SML0586, Sigma-Aldrich) (20 µM/well) for 24 h before the collection of supernatant and cells for supernatant cytokines and PCR as mentioned above. Notably, Lomeguatrib was dissolved in DMSO (Sigma-Aldrich) with the stock solution of 6.13 mM and was stored at −80 °C. Then, the stock solution was immediately diluted in DMEM before use to control the final DMSO concentration of less than 0.3%, according to a previous publication [122]. Additionally, several injury parameters, including cell-free DNA, malondialdehyde (MDA; a representative reactive oxygen species), and DNA break, were measured in *mgmt* null and *mgmt* control macrophages because of the influence of LPS in cell injury. As such, cell-free DNA in supernatants was measured by a PicoGreen assay kit (Invitrogen) following the manufacturer’s protocol [123]. For MDA, the activated macrophages were homogenized by the Ultra-Turrax homogenizer (IKA, Staufen, Germany) and centrifuged at 12,000× *g* for 15 min at 4 °C to separate the supernatant. Then, malondialdehyde (MDA) in the supernatant was measured by an MDA assay kit (colorimetric) (Abcam, Cambridge, UK) according to the manufacturer’s protocol for the intracellular reactive oxygen species (ROS) [124]. In parallel, immunofluorescence was used to determine DNA break. Accordingly, macrophages at 3 × 10^6^ cells were seeded on covered glass-bottomed 6-well plates before activation with LPS (100 ng/mL) or DMEM control for 24 h. Then, the cells were fixed with 4% paraformaldehyde in Tris Buffered Saline (TBS) for 15 min, permeabilized with 0.1% triton X-100, and washed three times in 1X TBS with 0.05% Tween-20. Fixed samples were blocked with 2% bovine serum albumin in 1X TBS for 1 h at room temperature and then incubated overnight at 4 °C with phospho-histone H2A.X (Ser139) (20E3) rabbit mAb (Cell signaling). Proteins were visualized using goat anti-mouse IgG H&L tagged Alexa Flour 488 (Abcam; ab150113) (green), and actin filaments have been labeled with DY-554 phalloidin (red) and snapped by confocal laser scanning microscope (CLSM, Zeiss, Germany) at 630× magnification in 10 randomly selected fields.

### 4.3. Animal and Animal Model

The animal protocol (017/2562) was approved by the Institutional Animal Care and Use Committee of the Faculty of Medicine, Chulalongkorn University, Bangkok, Thailand, according to the National Institutes of Health (NIH) criteria. For the proteomic analysis, macrophages were prepared from mouse long bones (8-week-old) of wild-type (WT) male C57BL/6 mice purchased from Nomura Siam (Pathumwan, Bangkok, Thailand). For other experiments, *mgmt*^flox/flox^ and LyM-Cre^Cre/Cre^ mice were purchased from RIKEN BRC Experimental Animal Division (Ibaraki, Japan) and cross-breed to produce *mgmt* littermate control (*mgmt*^fl/fl^; LysM-Cre^-/-^) or *mgmt* null (*mgmt*^fl/fl^; LysM-Cre^cre/-^) mice in F3 of the breeding protocol. Of note, the *mgmt*^flox/flox^ mice with the loxP sites were bred with LysM-Cre^Cre/Cre^ mice and the cre recombinase mice with the control of lysozyme M targeted deleted *mgmt* only in the myeloid cells (macrophages and neutrophils). The offsprings were (i) *mgmt*^flox/flox^ with no LysM-Cre (*mgmt*^fl/fl^; LysM-Cre^-/-^), categorized as the littermate controls or *mgmt* control, and (ii) *mgmt*^fl/fl^; LysM-Cre^cre/-^ (*mgmt* null) with positive for the Cre driver that lacks MGMT enzyme. Both groups of mice, including the conditional targeted Cre positive mice (*mgmt* null) and floxed/floxed littermate controls (*mgmt* control), were age- and gender-matched with the use of only 8–10 weeks-old male mice. To genotype these mice on the loxP sites insertion, the following primers were used; (i) LysM-cre primer; F: 5′-GAACGCACTGATTTCGACCA-3′, R: 5′-GCTAACCAGCGTTTTCGTTC-3′, (ii) *mgmt*-loxP primer F: 5′-TGGGCTTCAAATCAAGGAACAGAA-3′, R: 5′-AACTATCCTGCTCACTCTCTGTAG-3′, and (iii) Cre recombination (for Cre activity); F: 5′-GGTGTGGATCCCAAGAAATTGAAG-3′, R: 5′-TGTTCAAGAGTGACACACAGTCA-3′. The mice homozygous for the flox were selected and genotyped for the expression of LysM-Cre using the primers; F: 5′-CTTGGGCTGCCAGAATTCTC-3′; R: 5′-CCCAGAAATGCCAGATTACG-3′. For the test of MGMT inhibitor, only 8–10 weeks old male mice were used. Then, *mgmt* control, *mgmt* null, and WT C57BL/6 mice were used for the cecal ligation and puncture (CLP) surgery to induce sepsis or sham operation under isoflurane anesthesia following previous publications [125,126,127]. Briefly, the cecum was ligated at 10 cm from the cecal tip, punctured twice with a 21-gauge needle, and gently squeezed to express a small amount of fecal material before closing the abdominal wall (a midline abdominal incision) layer by layer with sutures (Nylon 4-0). After that, tramadol (25 mg/kg/dose) in 0.25 mL prewarmed normal saline solution (NSS) and imipenem/cilastatin (14 mg/kg/dose) in 0.2 mL NSS were subcutaneously administered in both frank area after surgery, and at 6 and 18 h post-CLP [6]. In the sepsis protocol without antibiotics, NSS in the same volume was used. In WT mice with CLP sepsis, an MGMT inhibitor (Lomeguatrib, SML0586, Sigma-Aldrich) at 1 mg/kg [43,128] mice in 3% dimethyl sulfoxide (DMSO) or DMSO alone (vehicle control) was subcutaneously administered at 15 min before surgery and at 6 h later (15 min before tramadol and the antibiotics). For sham-operated mice, the cecum was isolated and closed the abdomen by suturing without either ligated or punctured. 

### 4.4. Mouse Sample Analysis

For kidney and liver injury, serum creatinine and alanine transaminase, respectively, were measured by colorimetric method using QuantiChrom™ Creatinine Assay (BioAssay System, Hayward, CA, USA) and EnzyChrom Alanine Transaminase assay (EALT-100, BioAssay), respectively [129]. Serum cell-free DNA and LPS (endotoxin) were detected by Quanti PicoGreen assay (Sigma-Aldrich) and HEK-Blue LPS Detection Kit 2 (InvivoGen™, San Diego, CA, USA), respectively. Blood bacterial abundance (bacteremia) was evaluated using the direct spread of mouse blood onto blood agar plates (Oxoid, Hampshire, UK) in serial dilutions and incubating at 37 °C for 24 h before colony enumeration. Meanwhile, ELISA (Invitrogen, Carlsbad, CA, USA) was used to detect serum cytokines (TNF-α, IL-6, and IL-10). For kidney injury determination, the injury score was semi-quantitatively evaluated on Hematoxylin and eosin (H&E) staining in 4 mm thick paraffin-embedded slides at 200× magnification by the area of injury (tubular epithelial swelling, loss of brush border, vacuolar degeneration, necrotic tubules, cast formation, and desquamation) using the following score: 0, area < 5%; 1, area 5–10%; 2, area 10–25%; 3, area 25–50%; 4, area > 50%. The liver injury score was also measured through an area of hepatic injury, defined as congestion, degenerative cellular changes, cytoplasmic vacuolization, leukocyte infiltration, or cellular necrosis, in 200x magnification of 4 mm thick H&E-stained slides using ten randomly selected fields for each animal with the following scores per examination field: 0 for an area of damage of;10%, 1 for an area of damage of 10 to 25%; 2 for damage involving 25 to 50% of the area; 3 for damage involving 50 to 75% of the area, and 4 for 75 to 100% of the area being affected [130]. In parallel, for spleen apoptosis, spleens with 10% formalin fixation were stained by anti-active caspase three antibodies (Cell Signaling Technology, Beverly, MA, USA), using immunohistochemistry, and expressed in positive cells per high-power field (200× magnification) as previously published [66,131].

### 4.5. Statistical Analysis

All data were analyzed with GraphPad Prism6 and demonstrated in mean ± S.E.M (standard error). The one-way analysis of variance (ANOVA) with Tukey’s comparison test was used to compare among groups. Survival analysis was evaluated by the Log-rank test. A *p*-value less than 0.05 was considered significant. 

## 5. Conclusions

There were several key findings from our data. First, LPS-induced macrophage injury, as indicated by the proteomic analysis, increased ROS (MDA), cell-free DNA, and DNA break. Second, the importance of MGMT for immune responses against LPS in macrophages using siRNA and *mgmt* null cells. Third, the MGMT influence in sepsis was demonstrated by the reduced severity in CLP sepsis of *mgmt* null mice. Fourth, the importance of effective antibiotics during immune modification therapy in sepsis as sepsis protective effect of *mgmt* null mice was lost without antibiotics. As such, the proteomic analysis of LPS-activated macrophages demonstrated possible cell injury as indicated by reduced proteins on oxidative phosphorylation with the high proteasome, perhaps correlating with mitochondrial ROS and ubiquitination of the deformed proteins, respectively. The role of MGMT enzyme to counteract LPS-induced DNA methylation was demonstrated through the attenuation of LPS-induced cell injury (DNA break and cell-free DNA) and responses (reduced cytokine production) in *mgmt* null cells compared with LPS-activated control cells. Likewise, less severe sepsis in *mgmt* null mice (MGMT loss only in myeloid cells) in CLP sepsis with antibiotics indicates the importance of myeloid cells and antibiotics in sepsis. Nevertheless, MGMT blockage, an available drug in cancer therapy, attenuated only serum cytokines but not mortality in CLP with antibiotics, implying further drug administration and delivery development.

## Figures and Tables

**Figure 1 ijms-24-10175-f001:**
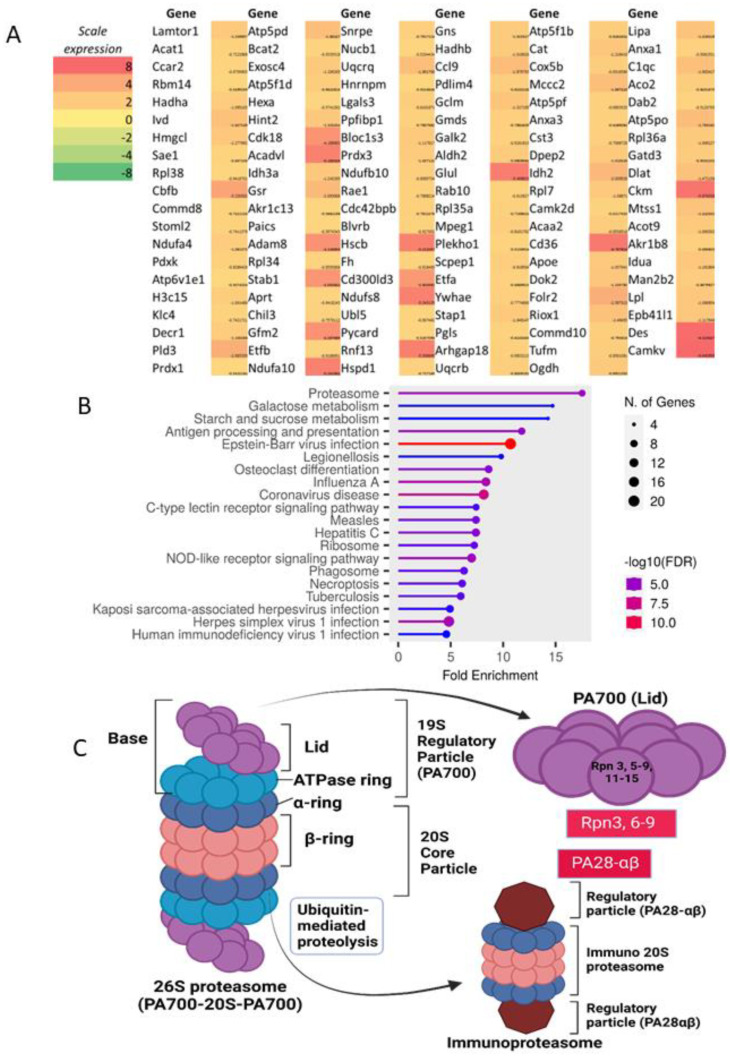
The proteome profiles of genes that generate peptides from wild-type bone marrow-derived macrophages after 24 h lipopolysaccharide (LPS) activation compared with media control as indicated by heatmap analysis (average value from 3 samples) using log2 of the count per million (TPM) of the upregulated genes (**A**) fold enrichment pathway of this list (**B**). KEGG analysis of the path (proteasome) (**C**) is demonstrated. The red boxes with the white letters in the gene’s name of KEGG pathway (**C**) indicate the analysis results. Macrophages were isolated from three different mice to perform triplicate experiments. Picture (**C**) is created by BioRender.com accessed on 15 March 2023.

**Figure 2 ijms-24-10175-f002:**
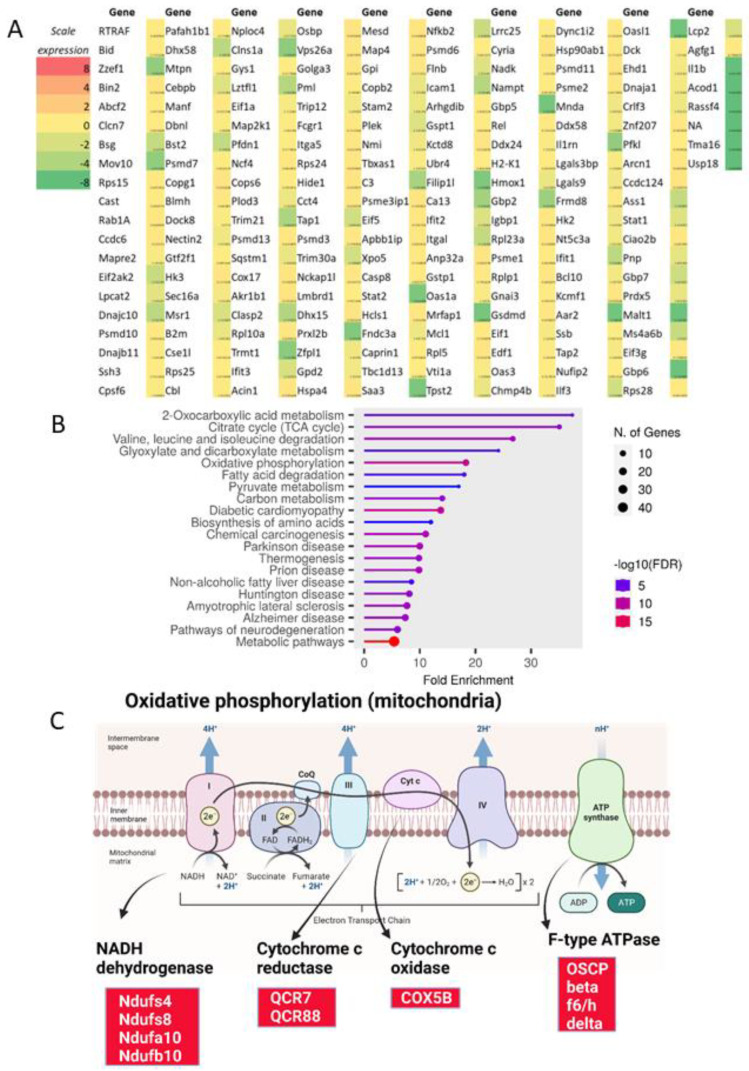
The proteome profiles of genes that generate peptides from wild-type bone marrow-derived macrophages after 24 h lipopolysaccharide (LPS) activation compared with media control as indicated by heatmap analysis (average value from 3 samples) using log2 of the count per million (TPM) of the downregulated genes (**A**) fold enrichment pathway of this list (**B**). In addition, KEGG analysis of the pathway (oxidative phosphorylation) (**C**) is demonstrated. The red boxes with the white letters in the gene’s name of KEGG pathway (**C**) indicate the analysis results. Macrophages were isolated from three different mice to perform triplicate experiments. Picture (**C**) is created by BioRender.com accessed on 15 March 2023.

**Figure 3 ijms-24-10175-f003:**
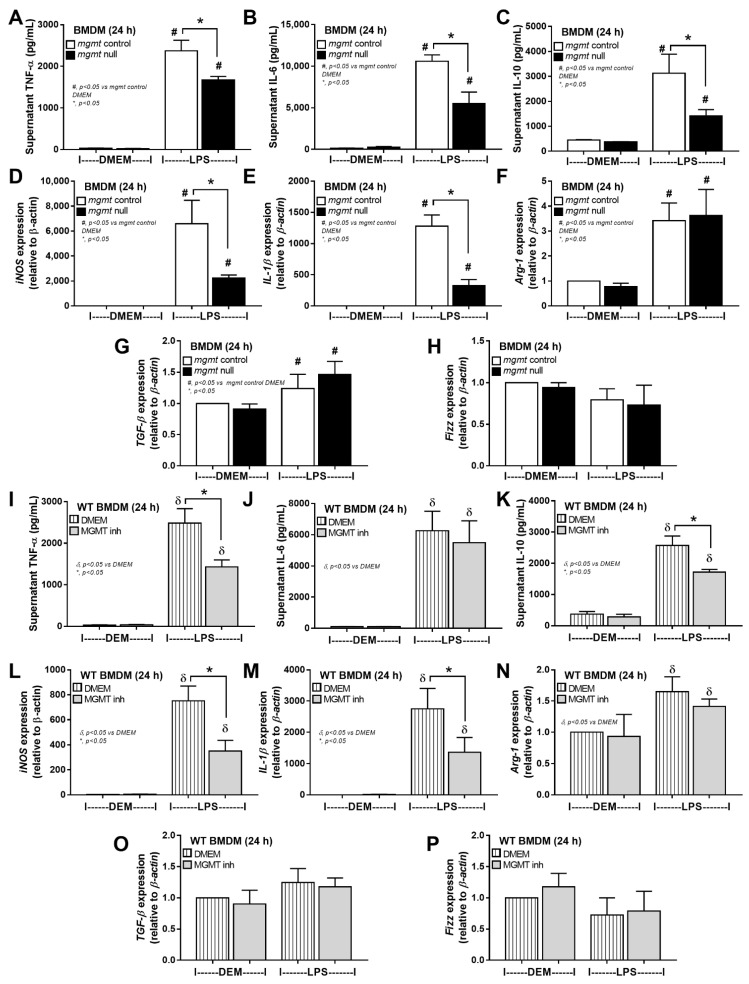
The characteristics of bone marrow-derived macrophages (BMDM) from mgmt control (mgmt^fl/fl^; LysM-Cre^-/-^) or mgmt null (mgmt^fl/fl^; LysM-Cre^cre/-^) mice at 24 h after activation by lipopolysaccharide (LPS) as indicated by supernatant cytokines (TNF-α, IL-6, and IL-10) (**A**–**C**), expression of pro-inflammatory genes of M1 polarization (iNOS and IL-1β) (**D**,**E**), and anti-inflammatory genes of M2 polarization (Arg-1, TGF-β, and Fizz-1) (**F**–**H**) are demonstrated. Parallelly, the characteristics of BMDM from wild-type mice at 24 h after activation by media (DMEM) or MGMT inhibitor (Lomeguatrib) with or without LPS as indicated by these parameters (**I**–**P**) is also demonstrated. Triplicated independent experiments were performed. Mean ± SEM with the one-way ANOVA followed by Tukey’s analysis was used. #, *p* ˂ 0.05 vs. *mgmt* control DMEM; δ, *p* ˂ 0.05 vs. control DMEM; *, *p* ˂ 0.05 between the indicated groups.

**Figure 4 ijms-24-10175-f004:**
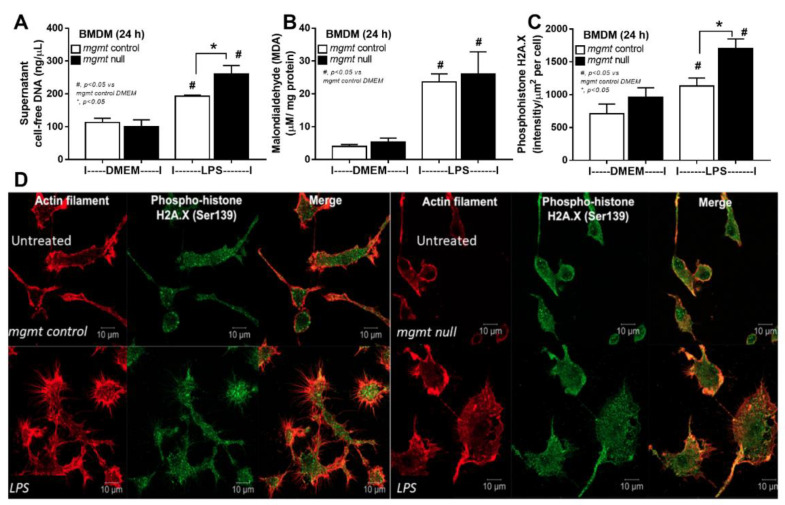
The characteristics of bone marrow-derived macrophages (BMDM) from mgmt control (mgmt^fl/fl^; LysM-Cre^-/-^) or mgmt null (mgmt^fl/fl^; LysM-Cre^cre/-^) mice at 24 h after activation by lipopolysaccharide (LPS) as indicated by supernatant cell-free DNA (**A**), malondialdehyde (MDA; a reactive stress molecule) (**B**), and immunofluorescent stained for DNA break (phospho-histone H2A.X; green color) and actin filament (red color) in intensity score and representative pictures (**C**,**D**) are demonstrated. Triplicated independent experiments were performed. Mean ± SEM with the one-way ANOVA followed by Tukey’s analysis was used. #, *p* ˂ 0.05 vs mgmt control DMEM; *, *p* ˂ 0.05 between the indicated groups.

**Figure 5 ijms-24-10175-f005:**
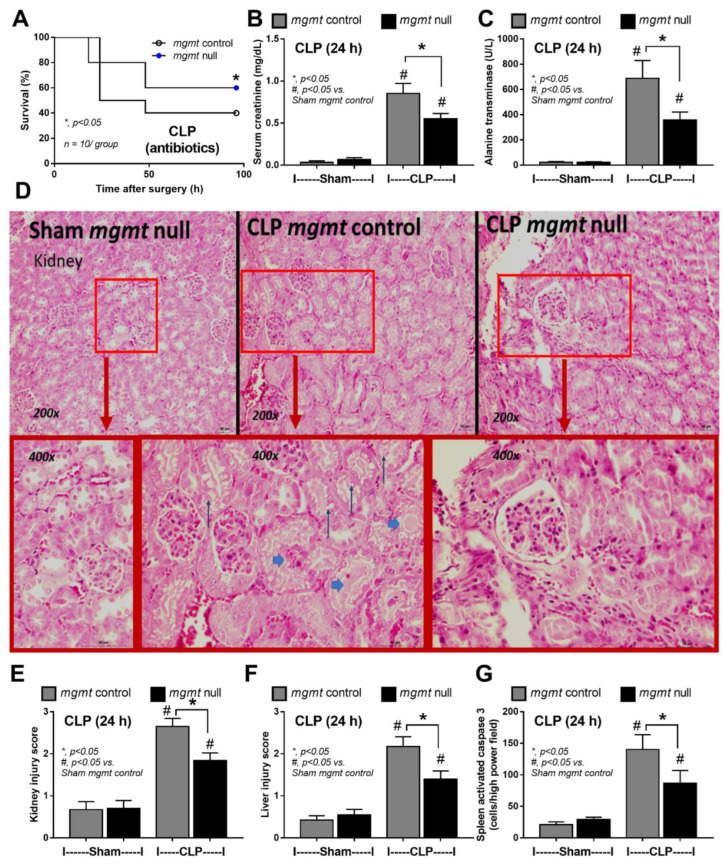
The characteristics of mice in mgmt control (mgmt^fl/fl^; LysM-Cre^-/-^) or mgmt null (mgmt^fl/fl^; LysM-Cre^cre/-^) group after cecal ligation and puncture (CLP) sepsis or sham control (Sham) surgery as indicated by survival analysis (**A**), and the parameters at 24 h-post surgery, including serum creatinine (**B**), alanine transaminase (**C**), renal histological score with representative pictures (**D**,**E**), liver injury score (**F**), and spleen apoptosis (activated caspase three immunohistochemistry) (**G**), is demonstrated (*n* = 10/group for A and *n* = 5–7/group for (**B**–**G**)). Mean ± SEM with the one-way ANOVA followed by Tukey’s analysis was used. #, *p* ˂ 0.05 vs. Sham mgmt control; *, *p* ˂ 0.05 between the indicated groups. Inset pictures, increased magnification; narrow arrows, tubular cell injury; broad arrows, renal tubular casts.

**Figure 6 ijms-24-10175-f006:**
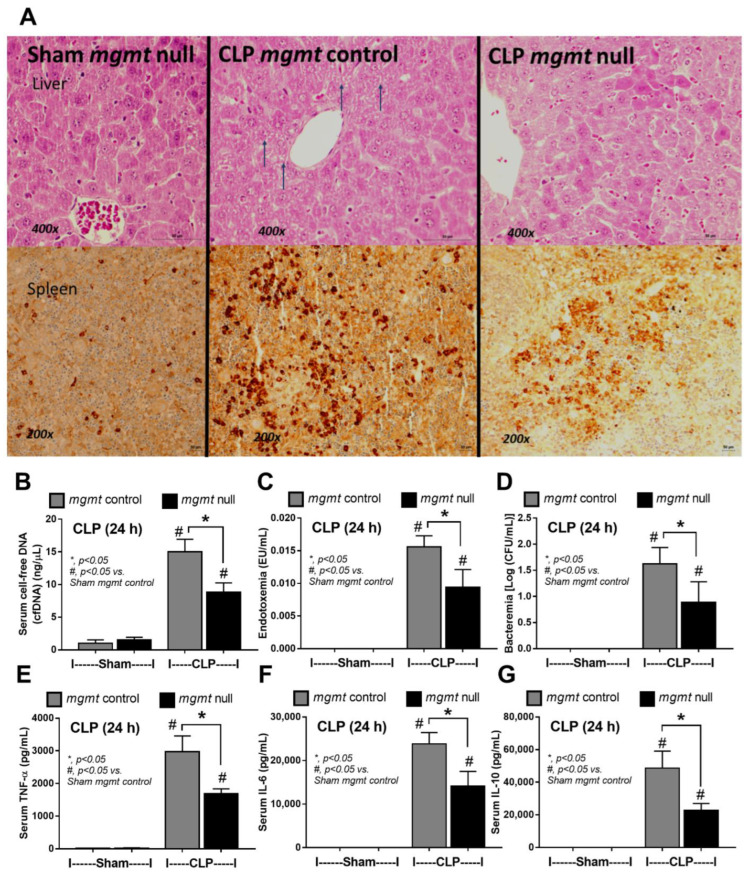
The characteristics of mice in mgmt control (mgmt^fl/fl^; LysM-Cre^-/-^) or mgmt null (mgmt^fl/fl^; LysM-Cre^cre/-^) group after 24 h of cecal ligation and puncture (CLP) sepsis (with antibiotic use) or sham control (Sham) surgery as indicated by representative pictures of the liver in Hematoxylin & eosin (H&E) staining and spleen activated caspase 3 (**A**), serum cell-free DNA (cf-DNA) (**B**), endotoxemia (**C**), bacteremia (**D**), and serum cytokines (TNF-α, IL-6, and IL-10) (**E**–**G**) are demonstrated (*n* = 5–7/group). Mean ± SEM with the one-way ANOVA followed by Tukey’s analysis was used. #, *p* ˂ 0.05 vs. Sham mgmt control; *, *p* ˂ 0.05 between the indicated groups. Narrow arrows, hepatocyte apoptosis.

**Figure 7 ijms-24-10175-f007:**
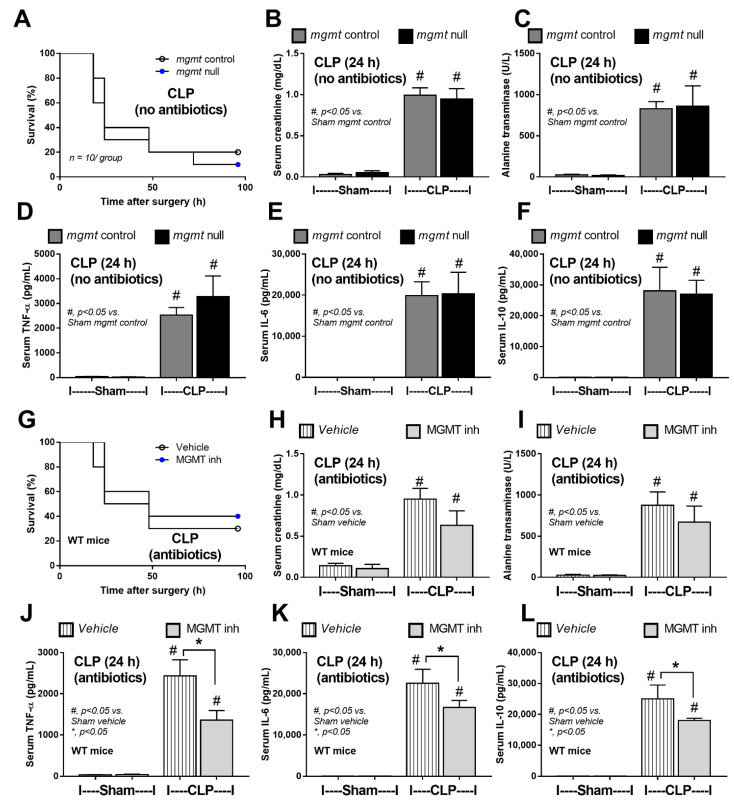
The characteristics of mice in mgmt control (mgmt^fl/fl^; LysM-Cre^-/-^) or mgmt null (mgmt^fl/fl^; LysM-Cre^cre/-^) group after 24 h of cecal ligation and puncture (CLP) sepsis (without antibiotic use) or sham control (Sham) surgery as indicated by survival (**A**), and the parameters at 24 h-post surgery, including serum creatinine (**B**), alanine transaminase (**C**), and serum cytokines (TNF-α, IL-6, and IL-10) (**D**–**F**) are demonstrated. In parallel, the characteristics of wild-type (WT) mice after CLP (with antibiotics) or Sham surgery with or without MGMT inhibitor (Lomeguatrib) as indicated by survival analysis (**G**), and the parameters at 24 h-post surgery, including serum creatinine (**H**), alanine transaminase (**I**), and serum cytokines (TNF-α, IL-6, and IL-10) (**J**–**L**), are demonstrated (*n* = 10/ group for survival studies and *n* = 5–7/group for others). Mean ± SEM with the one-way ANOVA followed by Tukey’s analysis was used. #, *p* ˂ 0.05 vs. Sham *mgmt* control; *, *p* ˂ 0.05 between the indicated groups.

**Table 1 ijms-24-10175-t001:** Lists of primers used in the study.

Name	Forward	Reverse
Inducible nitric oxide synthase *(iNOS)*	5′-ACCCACATCTGGCAGAATGAG-3′	5′-AGCCATGACCTTTCGCATTAG-3′
Interleukin-1β *(IL-1β)*	5′-GAAATGCCACCTTTTGACAGTG-3′	5′-TGGATGCTCTCATCAGGACAG-3′
Arginase-1 *(Arg-1)*	5′-CTTGGCTTGCTTCGGAACTC-3′	5′-GGAGAAGGCGTTTGCTTAGTT-3′
Resistin-like molecule-α1 *(Fizz-1)*	5′-GCCAGGTCCTGGAACCTTTC-3′	5′-GGAGCAGGGAGATGCAGATGA-3′
Transforming growth factor-β *(TGF-β)*	5′-CAGAGCTGCGCTTGCAGAG-3′	5′-GTCAGCAGCCGGTTACCAAG-3′
Beta-actin *(β-actin)*	5′-CGGTTCCGATGCCCTGAGGCTCTT-3′	5′-CGTCACACTTCATGATGGAATTGA-3′

## Data Availability

Not applicable.

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
