# Peer review of "Less Severe Polymicrobial Sepsis in Conditional mgmt-Deleted Mice Using LysM-Cre System, Impacts of DNA Methylation and MGMT Inhibitor in Sepsis"

_ijms, 2023, doi:10.3390/ijms241210175_

Round 1

Reviewer 1 Report

1) The authors state that MGTM enzyme inhibitors are already being used in cancer therapies. It would be beneficial to provide a more detailed explanation of their usage in these therapies to better understand their potential role in reducing sepsis.

2) The concept “Failure of DNA repair causes a reduction in macrophage functions, especially cytokine production and hyper-inflammatory responses, which possibly turns out to be a beneficial effect on sepsis that the authors present at the end of the paper, from line 378 to line 381, should also be mentioned in the introductory section of the paper.

3) The authors conducted a study on how inhibition of the enzyme MGTM may play a significant role during sepsis. To demonstrate this, they performed in vitro and in vivo experiments on mouse macrophages. Since the authors, as stated in the title of the studyLess severe polymicrobial sepsis in conditional mgmt-deleted mice using LysM-Cre system, impacts of DNA methylation in sepsisutilized the LysM-Cre recombination system, they should briefly mention the methodology as well.

Author Response

1. The authors state that MGTM enzyme inhibitors are already being used in cancer therapies. It would be beneficial to provide a more detailed explanation of their usage in these therapies to better understand their potential role in reducing sepsis.

ANS: We thank the reviewer for the comment and add more information in the new discussion as following “In cancer therapy, several alkylating agents destroy cancer cells through the induction of DNA methylation and some malignant cells resist these anti-malignant drugs partly by the increased production of DNA methylation enzymes, including MGMT, to counteract the DNA damage [90]. Hence, administration of MGMT inhibitors along with some alkylating agents enhance the anti-malignant effect of some cancers which using as an adjuvant therapy in some types of cancer [90].”.

2. The concept “Failure of DNA repair causes a reduction in macrophage functions, especially cytokine production and hyper-inflammatory responses, which possibly turns out to be a beneficial effect on sepsis” that the authors present at the end of the paper, from line 378 to line 381, should also be mentioned in the introductory section of the paper.

ANS: We thank the reviewer for the comment and add this hypothesis in the new introduction as following “Then, we hypothesized that failure of DNA repair causes a reduction in macro-phage functions, especially cytokine production, that might attenuate sepsis-induced hyper-inflammatory responses.”.

3. The authors conducted a study on how inhibition of the enzyme MGTM may play a significant role during sepsis. To demonstrate this, they performed in vitro and in vivo experiments on mouse macrophages. Since the authors, as stated in the title of the study “Less severe polymicrobial sepsis in conditional mgmt-deleted mice using LysM-Cre system, impacts of DNA methylation in sepsis” utilized the LysM-Cre recombination system, they should briefly mention the methodology as well.

ANS: We thank the reviewer for the comment and briefly mention the inhibitor in the new title as following “Less severe polymicrobial sepsis in conditional mgmt-deleted mice using LysM-Cre system, impacts of DNA methylation and MGMT inhibitor in sepsis”.

Reviewer 2 Report

The manuscript entitled ” Less severe polymicrobial sepsis in conditional mgmt-deleted mice using LysM-Cre system, impacts of DNA methylation in sepsis” by Kritsanawan Sae-khowet al focuses on the exploring the impact of mgmt on macrophage responses to LPS and cecal ligation and puncture (CLP) sepsis model using the conditional mgmt deletion mice with LysM-Cre system that selectively affected mgmt only in myeloid cells.

In conclusion, the authors of article noted an absence of mgmt in macrophages resulted in less severe CLP sepsis implying a possible influence of guanine DNA methylation and DNA repair in macrophages during sepsis.

The results obtained are significant for a better understanding of the mechanism of problem in question.

The validity of the results obtained and methods suggested is unquestionable.

The following improvements in the article would help other researchers to understand the significance of the findings obtained in this work.

1.            The key findings of the authors presented/noted are not enough clear. It is not evident the significance and perspective of findings obtained.

2.            The authors of the article were not explaining how the results obtained can be useful for treatment of pathologies.

3.            From the article is not clear how much animals were used for each group of experiments and number of cells/macrophages for peptides purification and Liquid chromatography–tandem mass spectrometry.

The English is fine enough. 

Author Response

The manuscript entitled ” Less severe polymicrobial sepsis in conditional mgmt-deleted mice using LysM-Cre system, impacts of DNA methylation in sepsis” by Kritsanawan Sae-khowet al focuses on the exploring the impact of mgmt on macrophage responses to LPS and cecal ligation and puncture (CLP) sepsis model using the conditional mgmt deletion mice with LysM-Cre system that selectively affected mgmt only in myeloid cells.

In conclusion, the authors of article noted an absence of mgmt in macrophages resulted in less severe CLP sepsis implying a possible influence of guanine DNA methylation and DNA repair in macrophages during sepsis. The results obtained are significant for a better understanding of the mechanism of problem in question. The validity of the results obtained and methods suggested is unquestionable. The following improvements in the article would help other researchers to understand the significance of the findings obtained in this work.

  1. The key findings of the authors presented/noted are not enough clear. It is not evident the significance and perspective of findings obtained.

ANS: We thank the reviewer for the comment and conclude the key finding in the new conclusion as following “There were several key findings from our data. First, LPS-induced macrophage injury as indicated by the proteomic analysis, increased ROS (MDA), cell-free DNA, and DNA break. Second, the importance of MGMT for immune responses against LPS in macrophages using siRNA and mgmt null cells. Third, the MGMT influence in sepsis was demonstrated by the reduced severity in CLP sepsis of mgmt null mice. Fourth, the importance of effective antibiotics during immune modification therapy in sepsis as sepsis protective effect of mgmt null mice was lost without antibiotics.”.    

  1. The authors of the article were not explaining how the results obtained can be useful for treatment of pathologies.

ANS: We thank the reviewer for the comment and conclude the key finding in the new discussion as following “Indeed, infiltration of immune cells in several organs during systemic inflammation, such as sepsis and auto-immune diseases, is one of the main pathogenesis of organ in-jury and the infiltration of cells with less pro-inflammatory activities by the MGMT interference here might induce less severe injury than infiltration by the very active immune cells [14, 69].”.

  1. From the article is not clear how much animals were used for each group of experiments and number of cells/macrophages for peptides purification and Liquid chromatography–tandem mass spectrometry.

ANS: We apologize for the unclear presentation and add this information in the figure legends (number of mice in each experimental group) and in the new method section. 
